# A Review of Trends in Corrosion-Resistant Structural Steels Research—From Theoretical Simulation to Data-Driven Directions

**DOI:** 10.3390/ma16093396

**Published:** 2023-04-26

**Authors:** Di Xu, Zibo Pei, Xiaojia Yang, Qing Li, Fan Zhang, Renzheng Zhu, Xuequn Cheng, Lingwei Ma

**Affiliations:** 1Institute for Advanced Materials and Technology, University of Science and Technology Beijing, Beijing 100083, Chinad202110654@ustb.xs.edu (R.Z.); mlw1215@ustb.edu.cn (L.M.); 2Shunde Graduate School, University of Science and Technology Beijing, Foshan 528399, China; 3Key State Laboratories, Wuhan Research Institute of Materials Protection, Wuhan 430030, China; 4National Materials Corrosion and Protection Scientific Data Center, University of Science and Technology Beijing, Beijing 100083, China

**Keywords:** data-driven, corrosion mechanism, material life, corrosion resistance modulation, modeling and simulation

## Abstract

This paper provides a review of models commonly used over the years in the study of microscopic models of material corrosion mechanisms, data mining methods and the corrosion-resistant performance control of structural steels. The virtual process of material corrosion is combined with experimental data to reflect the microscopic mechanism of material corrosion from a nano-scale to macro-scale, respectively. Data mining methods focus on predicting and modeling the corrosion rate and corrosion life of materials. Data-driven control of the corrosion resistance of structural steels is achieved through micro-alloying and organization structure control technology. Corrosion modeling has been used to assess the effects of alloying elements, grain size and organization purity on corrosion resistance, and to determine the contents of alloying elements.

## 1. Introduction

The material systems under study are becoming more and more complex with the development of the materials discipline, and accordingly it is more and more difficult to find regulars in complex material systems and to develop new materials. The main means of research and development of new materials are still based on the researcher’s scientific intuition and a large number of repeated trial-and-error experiments [1], such as outdoor exposure methods [2], electrochemical testing methods [3], indoor accelerated simulation methods [3], etc. These methods can directly or indirectly study the influence and the mechanism of a single variable on the corrosion failure process of structural steels. However, the structural steels corrosion failure process and time-varying regulars are complex, and the response signal is weak. It is difficult for traditional methods to accurately investigate the corrosion initiation mechanism of structural steels, and the amount of data obtained by these experimental methods is small and cannot accurately reflect the structural steels regular change over time [4]. Therefore, there is an urgent need to shorten the experimental period, but also to consider a variety of factors impacting the corrosion of structural steels data flow collection and processing methods to assist the development of corrosion-resistant structural steels.

In recent years, corrosion big data technology [5] has achieved rapid development, and has become an advanced means of efficient collection and analysis of corrosion process data under complex systems. Not only can corrosion data and various environmental factors data be collected continuously, but so can the multi-dimensional data stream for modeling and processing [6]. Therefore, the process of environmental corrosion and the main factors affecting of corrosion have only been recently understood. Because corrosion data can instantly reflect the influence of multi-dimensional variables on the output results, the amount of effective information obtained in a week is equivalent to that of a year or even a decade of data from the traditional method. As a result, the test cycle is greatly reduced and can be applied to control corrosion-resistant structural steels. Secondly, computational materials science [7] and integrated computational materials engineering [8] also propose the use of computational tools or the development of new computational tools to assist in the development of corrosion-resistant structural steels. The establishment of a digital corrosion database will provide a database for material corrosion research in order to achieve the storage of corrosion big data and the sharing of information. The integration of material corrosion mechanisms, corrosion big data, information science and computer technology to establish material corrosion models is the main research method for developing new corrosion-resistant structural steels.

In the paper, the research work on structural steels corrosion modeling is divided into three parts: the study of microscopic models of material corrosion mechanisms, the study of data mining models and the study of the performance control of corrosion-resistant structural steels. The main means of microscopic modeling of material corrosion mechanisms is to establish nano- to macroscopic two- and three-dimensional structural models, using first-principles, molecular dynamics, Monte Carlo simulations, cellular automata, finite elements and boundary elements to reproduce the material corrosion process in an order from small to large scales. Data mining model research involves the use of data mining tools for existing data sets to establish an abstract or figurative corrosion prediction mathematical model for the characteristics of the data set using the corresponding mathematical model. The main methods for the study of corrosion-resistant structural steels performance control are micro-alloying technology [9] and organization control technology [10]. Micro-alloying technology is the addition of corrosion-resistant alloying elements in structural steels to change the matrix corrosion resistance or the physical and chemical structure of the rust layer, thus improving the corrosion resistance of structural steels. Organization control technology involves heat treatment and other processes to refine the grain and make the organization structure pure. This paper provides an overview of the trends in the development of corrosion-resistant structural steels at home and abroad in recent years, and obtains corrosion mechanisms and regulars that cannot be obtained from traditional fragmentation data, which will become an important basis for the development of high-quality corrosion-resistant structural steels.

## 2. Microscopic Model of Corrosion Mechanism of Materials

The research on the corrosion mechanism of materials mainly reflects the corrosion process of materials by combining the virtual process of material corrosion with experimental data through computer software, so as to clarify the corrosion mechanism of materials. Density functional theory (DFT), molecular dynamics (MD), cellular automata, finite element simulation, boundary element simulation, gray correlation analysis and other methods were used to simulate the corrosion process of materials from nanoscale to macroscale, among which the Monte Carlo method can combine various theories to conduct multi-scale researches, as shown in Figure 1.

### 2.1. First-Principles

DFT describes the physical properties of the ground state of a system by the density of particles [11,12]. The calculations based on DFT are called first-principles, and software such as the Vienna Ab-initio Simulation Package (VASP) and Cambridge Sequential Total Energy Package (CASTEP) have been developed. In the field of corrosion, first-principles calculations are mainly focused on the study of the corrosion failure mechanisms of materials. In the past decade or so, first-principles calculations based on DFT have demonstrated powerful capabilities in studying the corrosion mechanisms [13,14,15,16], including the material surface properties, the interpretation and analysis of adsorption, diffusion, bonding and other behaviors presented in the corrosion processes on a nanoscale [17,18,19,20,21].

#### 2.1.1. Material Surface Model

Since the electrochemical corrosion processes are mainly related to the oxidation of O_2_ molecules and the formation of an electrical double layer by H_2_O molecules, most studies have been discussed with O_2_ and H_2_O molecules as the adsorption objects. In 2000, Graedel and Leygraf [22] found that it would take about 1 μs to form an oxide layer on the metal surfaces when they were exposed to the atmosphere, so the formation mechanism of the oxide layer cannot be investigated by conventional experimental methods. The decomposition and reconfiguration behavior of O_2_ on the Al (111) surface by first-principles was calculated and it was found that the differences in the coverage of the original adsorption sites would affect the type of adsorption configuration [23], while DFT calculations found that O_2_ molecules could only adsorb to the first FCC site on the Al (111) surface and could not penetrate to the subsurface [24]. The interface between metal and water also plays a crucial role in the corrosion process, so it is significant to study the bonding interaction between water and oxygen molecules on the material surface [25]. Although DFT calculations can effectively study the interactions between Al [26,27], Cu [28], Pt [29], etc., and metals and H_2_O molecules, the inability of the standard DFT method to represent the weak interactions between metal surfaces and H_2_O molecules makes the calculations difficult [30].

Moreover, it has been shown that the van der Waals dispersion forces should be taken into account when considering the interaction of metal surfaces with H_2_O molecules [31,32], and, in response to this view, Soria et al. [33] found that the van der Waals forces can only affect the adsorption energy magnitude and have no significant effect on the adsorption structure. Klimeš et al. [34] also improved the standard DFT method by including the van der Waals forces, which effectively corrected the errors for most systems that required the consideration of van der Waals interactions. In addition to the studies targeting the adsorption of O_2_ and H_2_O molecules, the adsorption behavior of SO_2_ molecules [35] and benzotriazole (BTA) [36] on the surface of the metal-passivated film was studied using a DFT method through adapting the algorithm. The results showed that the adsorption of SO_2_ molecules on the surface is greatly enhanced in the presence of O vacancies, further indicating the importance of the integrity of the passivated film on the material surface for corrosion resistance.

By studying the shielding effect of the material surface structure on corrosive molecules, it laterally reflects the corrosion resistance of the material or coating, and it is also important to study the corrosion resistance mechanism of materials. Based on DFT calculations, Li et al. [37] showed that BN films with more than two layers of structural integrity already possess a strong dielectric shielding effect, and this performance helps to protect metals from corrosion by stray currents. Kulmas et al. [38] investigated the effect of adding TiO_2_ on the performance of pure ZnO overlays by model calculations, and found that Ti atoms on the ZnO surface slightly reduced the energy band gap of ZnO, thus increasing the stability of the electrode and improving the corrosion resistance. 

#### 2.1.2. Internal Lattice Model

First-principles calculations can also simulate the internal structure of crystals and investigate the material failure process. Most of the studies focus on the development of hydrogen atom diffusion models to investigate the corrosion mechanism of materials by calculating the interfacial binding energy with the effect of the aggregated hydrogen atoms on the interfacial binding energy. Relevant calculations have been performed to demonstrate that hydrogen atoms readily occupy octahedral gaps within crystals [39,40,41,42]. The interaction of hydrogen atoms with 17 transition group metal surfaces was studied by the periodically self-consistent DFT-GGA (PW91) method [43]. It is found that the activation energy barrier of H diffusing inward on different surfaces is very different, even for the same metal. The diffusion energy barriers of hydrogen atoms are higher than the experimentally obtained values of the grain boundary energy barriers, indicating that the diffusion process of hydrogen atoms inside Er_2_O_3_ is controlled by the grain boundaries [44]. For the metal Al, H atom biasing causes a decrease in the grain boundary binding energy at each crystal plane [45]. Unlike the newly revealed mechanism, Fernandez et al. [46] investigated the diffusion of H atoms inside Wolfram(W) using a combination of DFT, transition state theory and a thermodynamic static model. By combining the temperature data obtained from the model with the macroscopic experimental results, it is effectively revealed that the deviation between the H diffusion coefficient measured by DFT and the experimental results is caused by the temperature difference and vacancy concentration.

In addition to diffusion modeling, some studies have taken the lattice itself as the object of study. The effect of cyclic loading on the crack formation threshold region of alumina was explored, and it was found that alumina lattices subjected to uniaxial tensile and unloading stresses can suffer from nanoscale fatigue, which in turn may affect the durability of ceramics [47]. Additionally, it proved further that the fracture energy can be obtained by calculating grain boundary energy and surface energy, which greatly simplified the characterization of fracture energy [48].

### 2.2. Molecular Dynamics

Since quantum chemical calculation methods based on first principles are usually only applicable to models where the system contains no more than 100 atoms or small molecules, and cannot support a large number of metal atoms and hundreds of solvent molecules at the same time, the molecular dynamics (MD) method is required to meet the needs of the simulation scale. MD uses the laws of endogenous dynamics to calculate and determine the shape transition of a system. At the beginning of the calculation, the MD method needs to determine the form of force between the atoms in the system, establish a mathematical model by means of Newtonian mechanics, and study the corresponding characteristics of particles by mathematical statistics. In recent years, due to the transition of the corrosion mechanism from a static to dynamic status, MD simulation has become an important research tool [49,50].

MD is frequently used to study phase interfaces [51,52]. For example, the size of the contact angle was simulated between water droplets and graphene coatings with different numbers of molecular layers, and the results showed that the water contact angle was independent of the number of graphene molecular layers, and that a single graphene molecular layer had good hydrophobicity [53]. The corrosion processes of oxygen molecules, chloride ions and metals have an important relationship with the types of metals. At the early stage of corrosion, it was found that the surface oxygen concentration plays a decisive role in the stability and corrosion resistance of the passivation film [54]. On the gold surface, the first layer of water molecules forms a network that is not connected to the gold surface, and the orientation of the water molecules is evenly distributed [55]. Prasanna et al. used molecular dynamics and quantum chemical parameters simulations to study the corrosion inhibition for soft-cast steel by the corrosion inhibitors olmesartan [56] and ketosulfone [57] in 1 mol dm^−3^ HCl. The experiment suggests that the inhibition efficiency of olmesartan increases with its increasing concentrations due to the adsorption at the temperature region of 303 K to 333 K. Even at a higher temperature of 333 K, the inhibitor molecules attain their stability towards the corrosion resistance of steel surfaces. The adsorption of olmesartan on steel surfaces is spontaneously found to include a mixture of physisorption and chemisorption. However, the inhibition efficiency of ketosulfone increases with an increase in concentration and with increase in temperature up to 313 K. The adsorption of the ketosulfone on a steel surface is predominately due to chemisorption and is spontaneous, which is confirmed by the activation parameters.

MD can also be used to simulate changes inside the material lattice to find the failure mechanism of materials. Beyerlein et al. [58] used MD modeling to confirm that radiation damage to materials is mainly caused by the stress concentration caused by the generation of vacancies in the material. The failure forms of galvanized iron products and copper products were simulated using MD. The simulation results showed that the onset of failure was accompanied by a corresponding stretching of the Zn atomic lattice [59], and the [110]/[100] interface of copper could more easily withstand the changes in temperature and load than other interface structures, but was less prone to the formation of failure behavior [60].

### 2.3. Monte Carlo Simulation at the Micro- and Nano-Scale

The Monte Carlo method is a process simulation method based on probabilistic and statistical theory methods. The basic idea is to set the corresponding base parameters to establish a probabilistic model or generate a stochastic process. Through multiple simulation operations on the model or process, the statistical distribution characteristics of the requested parameters are obtained, and the arithmetic mean is regarded as the approximate value of the requested solution. The Monte Carlo method can simulate the corrosion process in micro- and nano-scales by combining the first principles, molecular dynamics and other methods, and it can also set the simulation conditions as the pitting rate and other macro-scale parameters. The difference is that the use of Monte Carlo methods for micro- and nano-scale simulations is mostly aimed at studying the corrosion mechanism of materials, while macro-scale simulations are mainly used for service life prediction and safety assessment.

The Monte Carlo simulation methods can be used to model the effect of corrosion inhibitors on carbon steels by combination with MD and molecular mechanics. The effects of the hydrogen bond length and alkyl side chain of the inhibitor and the steel surface were obtained after calculation results and experimental results [61]. For instance, Sasikumar et al. [62] used the first-principles combined with the Monte Carlo simulation process to simulate the corrosion inhibition of alkylimidazolium tetrafluoroborate ions on carbon steel in an acidic environment at the atomic scale. Combining with the experimental verification, it is found that the corrosion inhibition efficiency of the compound is affected by the length of its alkyl side chain, and the corrosion inhibition effect is ranked as [C_10_MIM]^+^ [BF_4_]^−^ > [BDMIM]^+^ [BF_4_]^−^ > [EMIM]^+^ [BF_4_]^−^.

### 2.4. Cellular Automata

Cellular automata (CA) is an ideal physical system model, which is discrete in time and space and requires only a finite set of values for its physical states [63]. Usually, a probabilistic CA is used in the simulation of the corrosion process.

Cordoba-Torres et al. [64,65,66,67] first used the CA approach to model metal corrosion processes. The electrochemical reaction rate constants were replaced by probabilistic factors such as cellular automata evolution rules to simulate the anodic dissolution process of metals. The results show that there is a cellular islanding phenomenon in the simulation process, which is consistent with the actual mesoscopic morphology formed during the dissolution of the metal anode. In addition, the mesoscopic inhomogeneity is quantitatively analyzed based on the fractal theory. In addition, the CA method was used to simulate intergranular corrosion, pitting and uniform corrosion. The intergranular corrosion of the AA2024 aluminum alloy in Cl^−^ solution was simulated by the CA, and the results showed that both quantitative and qualitative results obtained by the model were in good agreement with the corrosion data measured by the experiments, which verified the scientific validity of the CA method to simulate the corrosion process [68]. The competing mechanisms of pitting and uniform corrosion were proposed from the perspective of corrosion kinetics and the relationship between corrosion kinetics and corrosion morphology by comparing the simulation and experimental results [69], which indicated that the growth of sub-pitting corrosion was controlled by the anode solution [70].

### 2.5. Finite Element Simulation and Boundary Element Simulation

The Finite Element Method [71] (FEM) is a modern computational method used to decompose the entire problem area, turning sub-regions into simple structural parts that can be solved easily. The application of finite element simulation based on mechanics in corrosion is mainly used to study the stress distribution around structural materials or pitting pits [72].

Turnbull et al. [73] studied the stress–strain distribution around the pitting pits on the surface of cylindrical specimens by finite element FEM simulation. They concluded that the stress at the bottom of the pits was the maximum and the strain at the shoulders of the pits was even higher. The development of the corrosion process in the pitting pits was simulated by reducing the number of meshes. Based on the cohesion model, the influence of hydrogen diffusion on corrosion was studied by FEM software [74,75], and the simulation data were consistent with the experimental results. Except for the diffusion of hydrogen atoms, the bias of hydrogen atoms also has a large effect on the sprouting and extension of intergranular cracks in metals [45]. At the same deformation displacement, the percentage of failure cells increased with decreased grain boundary binding energy, while under the same grain boundary binding energy, the percentage of failure units increased with the increase in displacement, which is an intuitive simulation and confirmation of the traditional hydrogen-induced intergranular cracking theory [76]. Based on the Abaqus method, the process of crack sprouting and extension was simulated through two-dimensional linear and secondary bonding cells [77], and a system was developed for predicting the critical internal stress value of high-strength steels with a given corrosion defect size through the FEM model. The system has been applied to assess pipeline safety in Canada.

Since the finite element method is too computationally intensive for three-dimensional problems, and for problems in infinite domains due to the inability to determine the boundary conditions and thus the accuracy of the solution, it is a good choice to apply the numerical calculation of the potential boundary element method in cathodic protection engineering. Li [78] established the potential distribution model of cathodic protection based on the boundary element method by discretely solving the Laplace equation of the mathematical model of cathodic protection. The results showed that the error of the boundary element method was controlled within 10% and the evaluation error was at 5%; so, the calculation results were satisfactory.

### 2.6. Grey Correlation Analysis

Grey system theory was proposed by Deng Julong in 1982, which was a new method to solve the problem of a small amount of data and poor information certainty. Grey system theory takes the “small sample” and “information-poor” uncertain system with partial information missing as the research object, and through mining the known information, the correct description of the behavior and evolution law of the system can be realized. Grey correlation analysis [79] is a data correlation analysis based on grey theory, which is used to find the key variables by mining the contribution weight of independent variables to the changes in dependent variables. At present, it is mainly applied to explore the correlation between various corrosion factors and material corrosion parameters under specific environments, and to find the key factors of the material corrosion mechanism.

Fu et al. [79] analyzed the correlation between oil and gas pipeline corrosion and environmental factor data by the grey correlation method, and found that the main factors causing oil and gas pipeline corrosion were sulfur-free corrosion and the erosion of oil and gas. Wang et al. [80] analyzed the influence of four atmospheric corrosion factors on the corrosion rate of distillation equipment in the distillation column by grey correlation quantification, and found that the salt content in oil and gas had the greatest influence on the corrosion of the equipment. Cao et al. [81] added more influence factors and analyzed the relationship between the corrosion rate of Q235 carbon steel accumulated in seven test stations in China, during a one-year period of atmospheric exposure corrosion tests (using weight loss method), and the data of 10 environmental factors affecting corrosion, and concluded that environmental factors contributed to the corrosion rate of carbon steel. However, the selection of these environmental factors is questionable in regard to whether the factors are correlated and repeatable, e.g., relative air humidity affects the time of condensation, while they are assumed to be uncorrelated during data mining. The limitation of grey correlation analysis is that it can only reflect the specific context of corrosion factors on the total weight of corrosion contribution. 

The material corrosion process from the nano to macroscopic scale simulation, using DFT, MD, CA, FEM, and the boundary element method of numerical calculation, showed through grey correlation analysis the contribution of environmental factors to corrosion. The results showed that grey correlation analysis cannot express the influence of changes in environmental factors on corrosion. Therefore, it is applicable to the analysis of the material corrosion mechanism in specific situations. 

## 3. Data Mining Methods for Corrosion Mechanism Research 

Material corrosion is a subject that relies on basic data. As human society enters the era of big data, the amount and complexity of corrosion-related data have greatly increased [82]. Viktor Mayer-Schönberger [83] pointed out four essential characteristics of big data (4V): volume, variety, velocity and value, which are also present in the corrosion-related data. Using data mining tools to effectively dig out hidden corrosion regulars from existing corrosion data and build a concrete or abstract prediction mathematical model is an important means to deal with this information. Corrosion data mining includes a variety of mining methods, which can be selected according to different types of corrosion data or prediction targets, as shown in Figure 2. At present, most corrosion prediction modeling research focuses on the prediction of the corrosion rate and corrosion life of materials, which are described in detail below.

### 3.1. Multiple Linear Regression Equation

Multiple linear regression refers to the method of establishing a mathematical model and making predictions by analyzing the correlation between two or more independent variables and one dependent variable. The general equation form is as follows.
(1)$$Y=a+b_{1}x_{1}+b_{2}x_{2}+b_{3}x_{3}+\ldots b_{n}x_{n}$$
where x_1_, x_2_, …, x_n_ are the independent variables, Y is the dependent variable, and the rest are unknown constants. Multiple linear equations were the first data mining methods used to predict the effect of environment on material corrosion.

In 1971, Haynie and Upham [84] proposed an atmospheric corrosion model for carbon steel with influencing factors such as SO_2_, exposure time and total oxides. However, due to the lack of comprehensive data on environmental factors, the fitting effect of this model was very poor. A Japanese research group used annual atmospheric environmental data (including temperature T, relative humidity RH, chloride ion Cl^−^, sulfur dioxide SO_2_ and precipitation rainfall) and corresponding exposure test data from seven locations to derive the carbon steel corrosion rate equation by multiple regression analysis [85].
(2)$$\begin{array}{l} corrosionrate\left(mdd\right)\\ =0.484\times T\left(℃\right)+0.701\times RH\left(\%\right)\\ +0.075\times \left[{Cl}^{-}\right]\left(\times 10^{-6}\right)+8.202\times \left[{SO}_{2}\right]\left(mdd\right)\\ -0.022\times Rainfall\left(\frac{mm}{mon}\right)-52.67 \end{array}$$
where *T* is the temperature, *RH* is the relative humidity, [Cl^−^] is the concentration of chloride ions, [SO_2_] is the concentration sulfur dioxide and Rainfall is the precipitation rainfall. This equation is important for the prediction of the corrosion rate of materials in an unknown atmospheric environment. After that, researchers tend to change the environmental conditions for the corrosion rate equation, for example, the corrosion rate of metals at different concentrations of hydrochloric and sulfuric acid solutions [86,87,88]. Since the multiple linear regression equation requires a linear relationship between the variables and a high linear regularity of the data itself, it has limited application in the field of material corrosion data.

### 3.2. Artificial Neural Networks

Artificial neural networks (ANN) is also the most widely used networks learning method in corrosion [89,90,91,92]. The ANN algorithm can adjust the connection strength between neural units in an adaptive manner through pre-set mathematical functions, and then can learn the knowledge from data samples. Compared with the multiple linear regression equation, ANN has the ability to handle nonlinear data, and also has a strong fault tolerance for data with noise interference. Therefore, in the modeling and prediction of corrosion data, ANN usually has better prediction accuracy compared with the multiple linear method. A typical neural networks model applied to corrosion is shown in Figure 3 [93]. The left input layer is fed with sample data of corrosion factors (such as humidity), and the right output layer is fed with sample data of the prediction target (e.g., first rust time), with a hidden layer in the middle for networks training.

Currently, ANN is mainly used for modeling corrosion data to adjust the input and output objects for material life prediction. In 1992, Smets and Bogaerts [94] used neural networks to predict the stress corrosion cracking of 304 austenitic stainless-steel samples in near-neutral Cl^−^ solution, which is one of the earliest applications of neural networks in corrosion research. The prediction accuracy of material corrosion resistance assessed by ANN is higher than that of the multiple linear regression model [93]. Shi et al. [90] used similar ANN models with input layers containing temperature, pH, electrochemical corrosion potential, conductivity and stress intensity factor (*Ki*) to predict the crack expansion rate of 600 alloy steels and 304 stainless steels due to stress corrosion, respectively, and the results showed that the errors of the predicted values were basically within the 95% confidence interval, and that all models showed good prediction accuracy. Many similar studies have been performed [95,96,97,98]. In order to substantially innovate the existing methods of corrosion data mining, ANN and various polarization curves are combined to predict the polarization curves and then the corrosion potential, polarization resistance and other parameters are derived, through which the corrosion resistance of materials can be judged [91]. The effect of time on corrosion kinetics is determined by changing the algorithm for the ANN model, using the current k moments as inputs to predict the output of the next moment [99]. Because of the large number of neural network parameters, there is a certain requirement for the sample size to establish the model, as otherwise there will be an overfitting problem.

### 3.3. Bayesian Networks

Bayesian networks are developed based on Bayes’ theorem, and they are graphical probabilistic networks based on probabilistic reasoning. Bayesian networks focus on mining the connections between variables and exploring the causality between corrupting factors. Bayesian networks are mostly used to model the carbonation corrosion process of reinforced concrete. The causal relationship between data was revealing, as the effects of temperature, humidity, Cl^−^ concentration, Cl^−^ diffusion coefficient, etc., on the carbonation coefficient k of reinforced concrete were explored. In addition, the time of corrosion onset was determined, while the corrosion onset time and the strength value of concrete together determined the corrosion current. This laterally confirmed the correctness of stress corrosion theory from the data perspective [100,101]. Because Bayesian networks are probabilistic statistical models, a certain amount of data was required to ensure the reliability of the model; so, it is impossible to conduct corrosion data mining in the case of little data and poor information.

### 3.4. Support Vector Machines and Support Vector Regression

Support vector machines (SVM) [102] is a supervised machine learning method built on the principle of structural risk minimization and the statistical learning theory VC dimensional theory. SVM aim to find the optimal hyperplane in a high-dimensional space containing all data equally spaced with different kinds of data sets, so that the type of unknown corrupted data can be predicted in an identifiable form. SVM have shown excellent performance in dealing with linear problems and high-dimensional pattern recognition problems, and have become an emerging tool for material data mining. They can even effectively use the “waste” data in the experiment, which is a revolution compared to the traditional research methods [103]. SVM can determine the type of corrosion through classifying data images by electrochemical noise [104]. At the same time, the optimized parameters were used to evaluate the corrosion state of weathering steel, and the results showed that the prediction accuracy of the SVM was higher than that of the ANN [105]. Qiu et al. [106] first proposed a SVM with recursive feature elimination to solve the atmospheric corrosion feature classification problem, but the model could not be fitted accurately for long-period prediction results due to an insufficient amount of sample data.

Support vector regression (SVR) is built on the theoretical basis of SVM, which aims to find the regression hyperplane with the minimum variance from the hyperplane for all sample points in the high-dimensional space. Unlike SVM, which favor the function of classifying discriminative predictions, SVR focuses more on building regression models for prediction. Based on the SVR method and back propagation (BP) neural networks, Wen [107] established a regression model of the corrosion rate of 3C steel in the marine environment under the influence of five environmental parameters (temperature, dissolved oxygen concentration, salinity, pH, redox potential), and compared the prediction accuracy of the two models, which also showed that the SVR method could predict high-dimensional corrosion data. In addition, the SVR method can be used to predict the corrosion inhibition efficiency of corrosion inhibitors [108]. The 19 amino acid and 20 benzimidazole derivatives corrosion inhibitors were studied and the structural parameters of the SVR model were improved to establish a nonlinear model of corrosion inhibitor efficiency; the results showed that the average error of this model in predicting the corrosion inhibition efficiency was as low as 1.48% [109].

### 3.5. Markov Chain

In 1906, the Russian mathematician A.A. Markov proposed the Markov chain, of which the evolution of the future state of things is only related to the present state as it is now known, but not to the past [110]. For example, Brownian motion can be considered as a Markov process. Multiple linear regression, ANN, SVM and SVR are applicable for mining multiple and discrete data. For example, for the multi-point etching pits under the same dimensional condition, each etching pit sample belongs to an independent sample and is not affected by other etching pit states. Compared with these data mining methods, the Markov process is more suitable for mining continuous and time-series type data. For example, for the same pitting pit depth over time, the pitting pit depth at the next moment is developed on the basis of the current depth considered to be related to the known current state, and is independent of the state of the pitting pit at the previous moment.

The current corrosion data mining objects through Markov chains are mainly focused on pitting corrosion datasets. Provan et al. [111] first used a non-singular Markov process to establish a pitting corrosion depth growth prediction model, and completed the first combination of Markov chains and material corrosion science research. A model for predicting the external pitting depth of X52 buried pipeline steels was established by a continuous time non-flush linear growth Markov process and it compared the prediction results with the experimental data. The experimental results showed that the prediction accuracy of the pitting corrosion depth is above 95%, but the prediction accuracy of the pitting corrosion distribution region is insufficient [112]. Based on Markov process analysis, researchers discussed the feasibility of jointly determining the service safety of equipment by multiple localized corrosion, and proposed the pipeline corrosion index (PCI) to measure the service safety of aging pipeline steel [113].

### 3.6. Monte Carlo Simulations at the Macroscopic Scale

The principle of the Monte Carlo simulation method has been described above, which is a means of data mining for process simulation by setting the corresponding base parameters to establish probabilistic models or generate stochastic processes. Unlike the previously introduced data mining tools, the material corrosion model established by the Monte Carlo simulation cannot adjust the input and output for flexible prediction. This part is mainly about macro-scale Monte Carlo simulation research, which can be applied to predict the pitting rate of the pipeline wall and to determine the safety of the pipeline service by establishing the distribution model of pitting depth or pitting rate in a certain time period.

Reigada and Sagues [114] first used Monte Carlo methods to discuss localized corrosion in 1994, and in the following year Wang et al. [115] studied the stress corrosion cracking behavior of Mn-Cr and Ni-Cr-Mo-V, thus opening a new era in the Monte Carlo simulation study of corrosion issues. Monte Carlo simulation is used to simulate the pitting rate, probability distribution of pitting depth and service safety reliability analysis of buried pipeline steels [116]. Additionally, it was found that the pitting growth is the main factor affecting the thinning of the pipeline steel’s inner wall; the pitting rate distribution was more consistent with F distribution than the Weibull and Gumbel distributions. The Markov process with non-uniform continuous time was used to establish the distribution model of the pitting depth of pipelines in future time, and verified that the prediction accuracy was high, which provided a new way to judge the reliability of pipelines in the future. For the inspection and maintenance problems within the pipeline at different stages, the authors divided the pipeline life cycle and maintenance decision into six and five states, respectively, and used the Markov model and Weibull distribution to establish the pipeline remaining life model to predict the future corrosion condition of the pipeline and the corresponding maintenance decision.

### 3.7. Grey Forecasting

As mentioned above, grey system theory is applicable to the study, but has the problems of few data, poor information and uncertainty. The core of grey prediction is to summarize multiple factors affecting the corrosion rate of materials over time, so the type of corrosion dataset must be in time series. Multiple linear regression, ANN, SVM, SVR, Markov chains and Monte Carlo simulation have a certain demand for the amount of data in the sample and cannot handle corrosion data with poor information. The grey prediction GM (1, 1) model only requires three to seven time-series corrosion data for mining and modeling, which can predict the change of corrosion-related parameters over time.

At present, GM (1, 1) mining studies on corrosion data are relatively homogeneous, most of which only predict corrosion rate by changing the specific context of the dataset mapping [117,118,119]. Based on the GM (1, 1) model, using grey theory to explore the corrosion of metals, the corrosion rate caused by oil on the bottom plate of stored petroleum containers was compared with the experimental data. The error interval was between 0.13% and 5.41%. It shows that the GM (1, 1) model has a high prediction accuracy for predicting the corrosion data [120,121,122].

## 4. Corrosion Resistance Performance Control by Data-Driven 

The evaluation of the corrosion resistance of structural steels is often verified by traditional experimental methods such as periodic immersion experiments, electrochemical experiments and outdoor exposure tests. However, the corrosion process and corrosion data obtained for structural steels have the characteristics of discontinuity and dispersion. This means that the traditional experimental methods cannot accurately assess the corrosion resistance of structural steels. To solve the above problems, the “big data technology research and development of corrosion-resistant steels” can effectively obtain a large number of environmental factors and the corrosion data of the structural steels field service, avoiding the problems of small amounts of experimental data and discrete data. This technology first uses a corrosion probe to capture the continuous and dynamic corrosion process data of structural steels and the corresponding environmental parameters, such as temperature, humidity, environmental pollutants, etc. It then uses big data mining technology to deeply explore the relationship between environmental parameters and structural steel composition, organization, structure and corrosion data, and finally regulates the corrosion resistance of structural steels. At present, the corrosion resistance of metal matrix and the stability of the rust layer are improved mainly by micro-alloying and organization modulation techniques. At present, the design concept of improving the corrosion resistance and rust layer stability of the metal matrix, and thus protecting the metal matrix, which is a high-quality corrosion-resistant material, is mainly realized through micro-alloying technology and organization structure control technology. Micro-alloying control technology refers to the addition of Mo, V, Nb, Cr, Ti and other elements in structural steels. After a large amount of data are obtained through the corrosion probe test and hanging test, using the Work function, Pearson correlation analysis and other methods to analyze the impact of the added alloying elements on the corrosion resistance of structural steels are used. The organization control technology is used to improve the corrosion resistance of structural steels by refining the grain size and purification of the organization through heat treatment. The effect of alloying elements or organization on the corrosion resistance of structural steels can be assessed using an electrical quantity diagram, clock diagram and F-index on the large amount of data obtained from both micro-alloying and organization control techniques. The performance control method of corrosion-resistant steels based on big data technology is shown in Figure 4. The development of a cross-scale theoretical model of material corrosion failure and the integration of data mining methods such as machine learning can significantly shorten the service evaluation cycle of structural steels and improve the accuracy of material life prediction.

### 4.1. Advances in Micro-Alloying Control Technology for Corrosion-Resistant Structural Steels

The most successful structural steels in the design and evaluation of corrosion resistance are weathering steels. In as early as 1900, European and American researchers discovered that Cu could improve the corrosion resistance of steels in the atmosphere [123]. In 1916, the American Society for Testing and Materials pioneered a standardized field corrosion test for atmospheric corrosion, which laid the foundation for corrosion evaluation techniques for the development of weathering steels. As a result, the United States, Germany, Britain and Japan began the research and development of weathering steels. The U.S. Steel Company first developed a high-strength corrosion-resistant copper-containing structural steels, i.e., Cor-ten steel, but it was very expensive [124]. From 1934 to 1958, after three large-scale field tests and studies, the theoretical basis of low-alloy weathering steels was basically laid, that is, the composition design principle of Cr-Ni-Cu-P and the concept of climate index. Climate index can evaluate the weathering performance of steels with different compositions, which is based on the accumulation of a large number of field corrosion test data. The formula is as follows.
I = 26.01(%Cu) + 3.88(%Ni) + 1.20(%Cr) + 1.49(%Si) + 17.28(%P) − 7.29(%Cu)(%Ni) − 9.10(%Ni)(%P) − 3.9(%Cu)(3)
where I is the climate index. The data in the formula are the mass percentage ratio of each component. The design requirements are I index ≥ 6.0 (Cu, P, Cr, Ni and Si, USA 1995). The formula was based on the design of a variety of ingredients and the statistical analysis of the results of numerous exposure experiments in the United States. The United States has been using this formula to determine the weathering properties of steels from the composition. However, the formula does not apply to Ni-series high-strength weathering steels. In the 1850s, the United States developed more economical weathering steels (A588) with a minimum yield strength of 350 MPa, which became the hallmark of high-strength weathering steels, and subsequently increased the yield strength to 690 MPa. The common series of weathering steels in the United States include the A242 series, A588 series, A606 series and A871 series. It should be pointed out that the I index does not take into account the types of inclusions, microstructure, surface state and environmental factors on the corrosion resistance performance [125].

From 1981 to 1993, Japan conducted exposure tests at 41 locations in its territory and found a large amount of corrosion resistance of weathering steels in salt conditions. This promoted the development of Ni-series weathering steels that could be used in coastal areas, and could gradually break the original restrictions on the use of weathering steels bridges. The weathering steels series commonly used in Japan include the SPA series and JIS SMA series. In 2003, the V index was proposed (≥0.9, adding C, Mn, S, Ti and Mo). Here, V is the indicator of the corrosion resistance of structural steels.
V = 1/{(1.0 − 0.16(%C)) (1.05 − 0.05(%Si)) (1.04 − 0.016(%Mn)) (1.0 − 0.5(%P)) (1.0 + 1.9(%S)) (1.0 − 0.10(%Cu)) (1.0 − 0.12(%Ni)) (1.0 − 0.3(%Mo)) (1.0 − 1.7 (%Ti))}(4)

In Europe, weathering steels also appeared early, and the UK has general provisions for weathering steels in British Standard European Norm (BS EN) 10155. In UK bridge engineering, S355J2G1W is the most commonly used weathering steel and its mechanical properties are similar to those of S355, as specified in BS EN 10025.

China began to develop weathering steels in 1950. In 1965, 09MnCuPTi weathering steels were trial-produced for the first time in China, and a number of new steel grades containing Cu, P, Ti, RE and other elements were also developed in combination with domestic resources, such as 08CuPVRE series, 09CuPTi series, 09MnNb, etc. [126,127]. Since 1983, a 20-year-long data accumulation work has been carried out. So far, it has integrated the material corrosion data production resources consisting of more than 30 national field stations in China and a large number of overseas cooperative observation sites. The aim is to develop various types of corrosion-resistant materials, especially corrosion-resistant steels. In recent years, with the rapid development of China’s steel industry, high-strength weathering steels also have experienced great development, and new high-strength weathering steels have emerged one after another. At present, the commonly used weathering steels in China are 09CuPCrNi, with a yield strength of not less than 345 MPa, a tensile strength of not less than 480 MPa, elongation of not less than 22%, corrosion resistance for ordinary carbon steels being two to eight times greater, and their use in trucks has a long history. With the development of Chinese steelmaking technology, the steel yield strength of 600 MPa or more has been available in the steelworks. In recent years, CrNiCuP and 3Ni-series weathering steels of 690 MPa grade have been successfully used in cross-sea bridges. According to different types of structures, high-strength weathering steels and welding weathering steels have achieved positive improvement of strength, toughness, corrosion resistance and welding performance by regulating alloying elements, such as Nb, V, Ti, Sb, Cr, Mo, Sn, Ca and rare earth elements.

Ni [128] and Cr [129] alloying elements can improve the corrosion potential of steels due to their stable thermodynamic properties, so as to improve corrosion resistance. The Cr [130] element can form smaller grains in the rust layer of the steel, making it dense and cation-selective, and the rust layer exhibits a significant inhibitory effect against Cl^−^ intrusion. Meanwhile, Ca [131] and Rare Element (RE) [132] improve the size and properties of inclusions and promote the generation of α-FeOOH to reduce the aggressiveness. Both Nb and V can effectively precipitate high-energy hydrogen traps to improve the hydrogen resistance of steel. In addition, refined grains can reduce interfacial hydrogen concentration by increasing grain boundary area [133]. Sb reduces various defects in the rust layer, especially the stability of the outer rust layer, and promotes the conversion of γ-FeOOH to α-FeOOH, thus improving the corrosion resistance of the steels [134,135,136]. There have been many studies on the use of microalloying techniques to obtain corrosion-resistant structural steels, but the data obtained from these studies are severely fragmented and the test results have large errors. Therefore, it is very important to establish a big data accumulation method for the corrosion of corrosion-resistant structural steels and to form a cross-scale research technology based on macroscopic corrosion behavior.

Yang et al. [137] combined corrosion big data and machine learning to analyze the differences in the effects of microelements Sn and Sb on the corrosion resistance of structural steels. Sn and Sb inhibit the corrosion behavior of structural steels in general, and structural steels containing 0.10% Sb have the best corrosion resistance. Jia [138] determined the influence of regular factors and the mechanisms of elements such as Ni, Mn and Cu on the stress corrosion resistance of 690 MPa steel based on the study of electro-couple big data technology. Pan [139] combined corrosion big data methods and machine learning methods to determine the influence of regular factors and the mechanism of the stress corrosion behavior of duplex stainless steel and its weld organization in the seawater environment. It was found that it can significantly improve the corrosion cracking resistance of offshore steel by inhibiting microscopic hydrogen damage and local anodic dissolution processes through compounding with micro-alloying corrosion-resistant elements such as Nb, Sb, Cu and Ni [140,141,142,143].

Yang et al. [144] evaluated the influence of the Cr element on the corrosion behavior of weathering steels, and the effect of the Cr element on the corrosion evolution behavior of weathering steels was evaluated by combining big data techniques with in situ corrosion specimens. To quantitatively describe the effect of Cr, an evaluation index based on mass loss was proposed with the expression shown in Equation (5), where *Q_B_* is the cumulative corrosion of the base material and *Q_Ele_* is the cumulative corrosion of the material when corrosion-resistant elements are added. *F* is the evaluation index with a value range of [−1, 1]. When *F* ∈ (0, 1), it indicates that the addition of elements in the structural steels is beneficial for improving the steel’s corrosion resistance. The closer the *F* value is to one, the better the improvement effect. When *F* ∈ [−1, 0], it indicates that the addition of the element in the structural steels is not beneficial or is even harmful for corrosion resistance. The closer the *F* value is to −1, the greater the effect of the element on the reduction in corrosion resistance. Additionally, when *F* = 0, it shows that the addition of this element has no effect on the corrosion resistance of structural steels. Atmospheric corrosion monitoring sensor technology is shown in Figure 5. Figure 5c shows a statistical analysis of the pitting corrosion of structural steels after 6 months of exposure to sunlight, and after the addition of different contents of Cr elements. As the Cr content increases, the slope of the linear fitting gradually decreases and the overall pitting developed in the direction of depth does too, indicating that the increase in Cr content in the structural steels promotes the development of pitting. Figure 5d shows the evolution of the *F*-index after the addition of different contents of Cr elements. The *F*-index is greater than 1, indicating that the addition of Cr is beneficial to the improvement of the uniform corrosion resistance of the structural steels. Combining the effects of Cr content on corrosion resistance in Figure 5c,d, the *F*-index can be used as a competing mechanism for the beneficial effects of Cr elements on uniform corrosion and the harmful effects on pitting resistance.
(5)$$F=\frac{Q_{B}-Q_{Ele}}{Q_{B}+Q_{Ele}}$$

However, the amount of alloying elements in structural steels still requires a large number of experiments to explore it further, increasing the difficulty and cost of the experiment. Therefore, Sun introduced the work function, which indicated the minimum energy required for electrons to escape from a solid surface, i.e., the smaller the work function, the greater the possibility of it losing electrons and the more likely corrosion will occur [145,146,147]. Based on the first-principles calculation of density functional theory, the influence of Cr, Mo and Sn elements on the surface work function was studied by MEDEA-VASP 5.4 software. A 6-layer supercell Fe surface model with a 15 Å vacuum layer is used during calculation. The (110) surface with the lowest surface energy of the bcc-Fe structure was selected [148], and the surface optimization was performed after selecting the same Fe atom position in the Fe surface layer for the doping of Cr, Mo and Sn atoms. The results showed that the addition of Sn to the steel is more helpful in improving the corrosion resistance of the steel itself [149]. Table 1 summarizes the test methods and machine learning methods used to study the corrosion resistance of structural steels reported in recent studies. Among these studies, the test methods were similar, while the machine learning methods differed due to the type of data.

To more accurately obtain the contents of three alloying elements Cr, Mo, and Sn in the structural steels, a dual evaluation model of corrosion rate and pit depth was constructed by Pearson correlation analysis based on the existing corrosion data of structural steels in the tropical marine atmosphere. The ability of the evaluation model and the reliability of the accelerated indoor tests were verified by combining the outdoor exposure test data. Combining the corrosion rate and pit depth model with the corrosion resistance range of Cr, Mo and Sn, it can be inferred that the overall corrosion resistance of structural steels is optimal when the contents of Cr, Mo and Sn are 2.5 wt.%, 0.25 wt.% and 0.22 wt.%, respectively, in the tropical marine atmosphere.

### 4.2. Organization Control Technology

Organization control technology has been widely used to improve the mechanical properties of structural steels. The main purpose of organization structure control is to refine the grain and purify the organization structure, which not only improves the strength, elongation and toughness of steel, but also improves the corrosion resistance of the structural steel [150].

From the Hall–Petch relationship, it is clear that grain refinement can improve the mechanical properties. The effect of grain refinement on corrosion resistance requires a lot of work to prove the relationship between the two. The early view was that the uniform corrosion rate of metallic materials and alloys increases with decreasing grain size. However, Ralston et al. [151] reviewed the effect of grain size on the corrosion resistance of materials and concluded that in an activated environment, grain refinement leads to a decrease in corrosion resistance, while in a passivated environment, grain refinement leads to an increase in corrosion resistance. Wang et al. [152,153] studied the corrosion behavior of bulk nanocrystalline iron and coarse crystalline iron in acidic solutions, among which bulk nanocrystalline iron has a better acidic solution. Liu et al. [154] further stated that in the activated environment, if the corrosion products are soluble, grain refinement will lead to accelerated corrosion. If the corrosion products are insoluble, grain refinement will improve the corrosion resistance. In a passivation environment, grain refinement contributes to the formation of a dense film, which affects the semiconductor properties. However, the results of experiments showed that the grain size mainly affected the initial stage of corrosion, and the fine grain size was more helpful in the formation of the surface protective rust layer, but the effect on the improvement in corrosion resistance was not obvious when the grain size was refined to a certain degree [155]. The rust layer was gradually stabilized in the later stage of corrosion, and the effect of grain size difference in the steel matrix on the corrosion resistance of steels was no longer obvious. Therefore, how the grain size affects material corrosion is of great research importance.

The purification of structural steel organization is also an effective means to improve corrosion resistance and mechanical properties. Because of the excellent mechanical properties of ultra-low carbon bainite organization, structural steels with this organization as the main phase have become a trend in the development of new high-strength and corrosion-resistant structural steels. The structure of low temperature bainite in structural steels after optimization mainly consists of carbon free bainite ferrite and residual austenite film, which not only improves the corrosion resistance, but also improves the strength [156,157]. In addition, the addition of specific elements in structural steels can optimize the structure, as with the increase in Mo content, the phase transition temperature decreases, the bainite zone expands, and the corrosion resistance is significantly improved [158]. Secondly, the surface of the sample containing only a ferritic structure tends to form uniform corrosion product films with fewer cracks, and its corrosion resistance is better than that of the mixed ferritic and pearlite structures [159]. There have been many studies on the corrosion resistance of structural steels that have focused on regulating the structure. However, due to the lack of data, it is still unclear how the relevant organization structure affects the corrosion resistance of structural steels.

**Table 1 materials-16-03396-t001:** The testing methods and machine learning methods.

Reference		Test Methods	Machine Learning
[137]	Sn/Sb	SEM, EDS, XRD, EBSD, XPS, SECM, Raman, Electrochemical test, SSRT, Periodic infiltration simulation acceleration experiment, Corrosion big data detectors,	RF
[138]	Ni/Mn/Cu	SEM, XRD, XPS, TEM, SAED, EBSD, Electrochemical test, SSRT, SAED, Periodic infiltration simulation acceleration experiment, Corrosion big data detectors, Hydrogen filling experiment,	GBDT
[143]	Nb/Cu/Sb	SEM, EBSD, TEM, XRD, XPS, Electrochemical test, Axial stress corrosion fatigue test	Work Function, PCC, SVC, SVR, LC, RF, MLP, KNN
[149]	Cr/Sn/Mo/Grain size	SEM, EBSD, EDS, AFM, EDS, XRD, XPS, TEM, CLSM, Electrochemical test, SSRT, Periodic infiltration simulation acceleration experiment, Corrosion big data detectors,	PCC, Work Function
[160]	Cr/Sn/Mo/M-A organization	SEM, EDAX, XRD, XPS, TEM, AFM, Periodic infiltration simulation acceleration experiment, Corrosion big data detectors,	ANN, SVM, RF, DNN

SEM—Scanning Electron Microscope; EBSD—Electron Backscattered Diffraction; EDS—Energy Dispersive Spectrometer; AFM—Atomic Force Microscope; XRD—X-Ray Diffraction; XPS—X-Ray Photoelectron Spectroscopy; SAED—Selected Area Electron Diffraction; TEM—Transmission Electron Microscope; CLSM—Confocal Laser Scanning Microscope; SECM—Scanning Electrochemical Microscopy; AFM—Atomic Force Microscope; EDAX—Energy Dispersive X-ray Analysis; SSRT—Slow Strain Rate Tension; RF—Random Forecast; PCC—Pearson Correlation Coefficient; LC—Logistic Classification; SVR—Support Vector Regression; SVM—Support Vector Machines; MLP—Multilayer Perceptron; KNN—k-Nearest Neighbor; DNN—Deep Neural Network; GBDT—Gradient Boosting Decision Tree.

The influence of organization structure on the corrosion resistance of structural steels is usually determined by conventional electrochemical methods, while the influence of organization on the corrosion resistance of materials is still fuzzy due to the uncertainty of the hanging method and the environment. Big data techniques are superior to traditional electrochemical techniques in identifying the differences in corrosion resistance between materials with small differences in corrosion resistance or with time-varying corrosion resistance. The corrosion clock diagram (Figure 6a_1_–f_1_, blue color indicates low corrosion rate, red color indicates a higher corrosion rate.) and the corrosion cumulative electrical quantity diagram (Figure 6a_2_–f_2_) were used to investigate the effects of austenite grain size, bainite lath thickness, and cathode to anode phase ratios on the corrosion resistance of structural steels (Figure 6 data from the author’s thesis). It is found that the corrosion resistance of structural steels can be improved by refining the original austenite grain size and the bainitic ferrite sub-crystalline grain size, as well as by reducing the content of M-A group elements. The combination of traditional experiments and big data experiments not only allows for corrosion regulation using different organizations at different times, but also determines the ranking of the corrosion rates using different organizations [160]. For instance, the corrosion resistance of four types of organizations in the heat-affected zone is high in the initial stage of corrosion, namely, coarse crystalline organization, fine crystalline organization, two-phase organization (coarse and fine crystalline) and matrix. With the extension of corrosion time, the corrosion rate of coarse crystal organization is always the highest, and the corrosion rate of fine crystal organization is the lowest; the matrix shows a high corrosion rate at the initial corrosion stage, but decreases rapidly at the later stage. The final corrosion result is coarse crystal organization > two-phase organization > matrix > fine crystal organization [149].

In the further study of the influence of alloying elements and organizational structure on the corrosion behavior of structural steels, it was concluded that the corrosion big data method can accurately identify the influence of various factors, such as small changes in various alloying elements and small differences in microstructure, on the corrosion resistance of structural steels, which is an effective and promising corrosion research method [160].

## 5. Conclusions

In recent years, the development of corrosion-resistant structural steels is moving from macroscopic to nano-scale characterization directions, from qualitative to precise quantitative directions, from long-term to rapid test directions, from fragmented data to big data directions, and from theoretical simulation to data-driven directions. This paper presents the study of a microscopic model of the material corrosion mechanism, the study of a data mining model, and the study of the performance control of corrosion-resistant structural steels, and new insights into the corrosion mechanisms and patterns of structural steels have been gained: (1) Combining corrosion test data with virtual processes through computer technology is useful for the study of structural steels corrosion mechanisms and patterns. (2) The data mining model is mainly used to predict the corrosion rate and corrosion life of structural steels. (3) The Work function and F-index can be used to assess the effect of alloying elements on the corrosion resistance of materials. (4) The Pearson correlation analysis method can be used to construct a dual evaluation model of corrosion rate and pit depth, and combined with the range of corrosion resistance of alloying elements, it can be inferred that alloying elements at a certain level can help structural steels to achieve excellent corrosion resistance. (5) The methods of corrosion clock diagrams and cumulative corrosion power diagrams allow for the magnitude of corrosion rates for different grain sizes to be ranked at different times. (6) The lower the M-A organization content in structural steels, the better the corrosion resistance.

Research work regarding material corrosion modeling has provided new approaches to the study of scientific problems such as material corrosion mechanisms, life assessment and new corrosion-resistant materials. However, the corrosion modeling of structural steels still needs to be combined with laboratory micro/macro characterization. Therefore, how to use material corrosion modeling to partially replace or completely replace field tests in the development of corrosion-resistant structural steels will become an important part of the work. Secondly, the existing structural steels corrosion data are not perfect, and understanding how to standardize and perfect the corrosion data is an important part of the corrosion database construction work. Thirdly, more research is needed on how to dig out corrosion big data to maximize function. Additionally, the research and development of high-quality corrosion-resistant structural steels represents a long-term goal; proficiency in material corrosion mechanisms and the in-depth study of material corrosion models will provide more convenient ways to develop corrosion-resistant materials.

## Figures and Tables

**Figure 1 materials-16-03396-f001:**
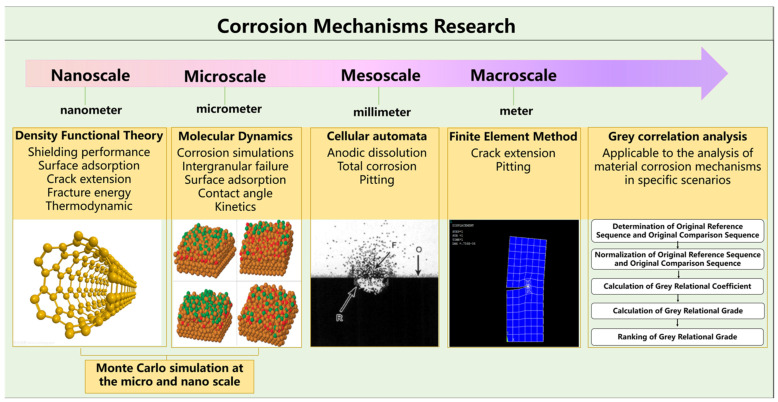
Microscopic model of material corrosion mechanism, R present the corrosion products, F present the Fe^2+^, O present the OH^−^.

**Figure 2 materials-16-03396-f002:**
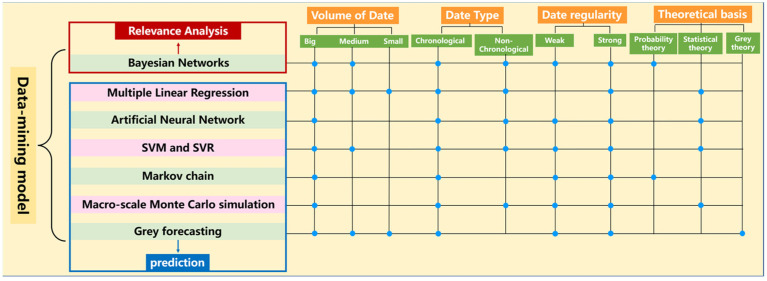
Data-mining model, blue circles present the features possessed by the data mining model.

**Figure 3 materials-16-03396-f003:**
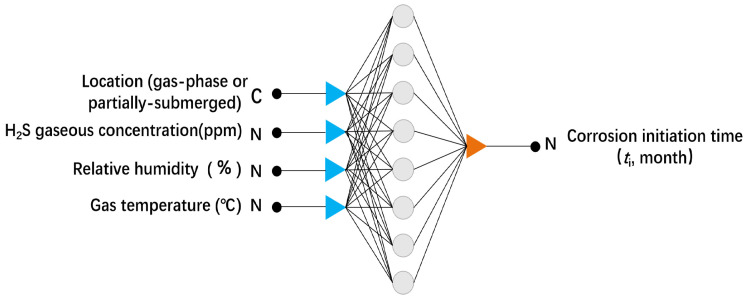
The network architecture of the ANN model for the prediction of corrosion initiation time(the circles presents neurons in the hidden layer.) [93].

**Figure 4 materials-16-03396-f004:**
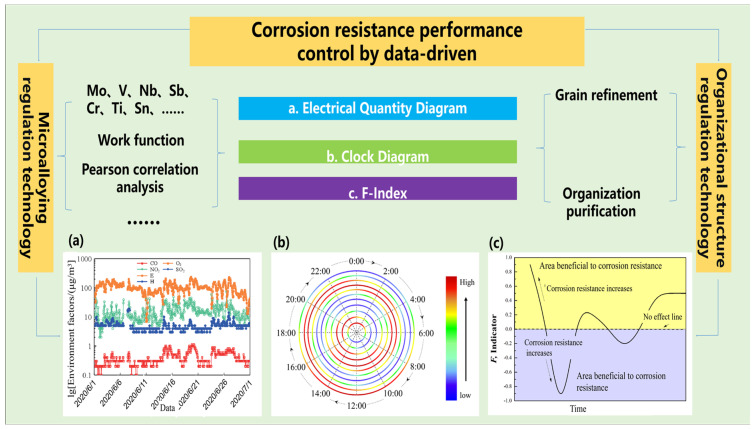
Performance control of corrosion-resistant structural steels based on big data, (**a**) electrical quantity diagram, (**b**) clock diagram, (**c**) F-index.

**Figure 5 materials-16-03396-f005:**
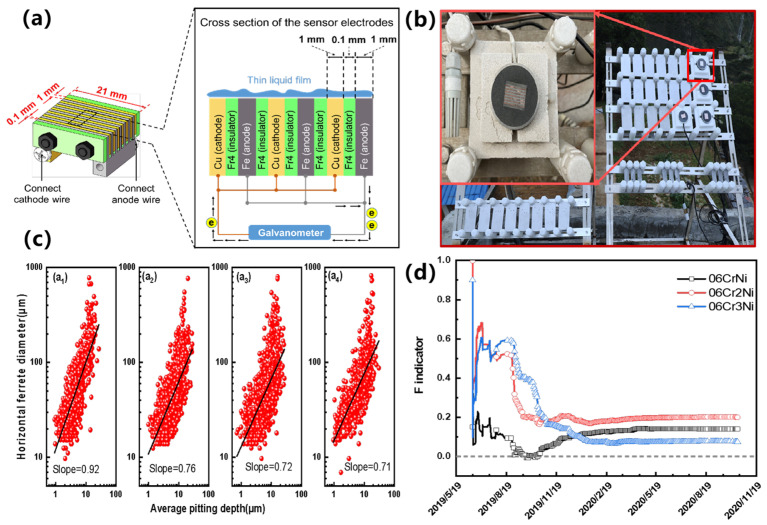
Atmospheric corrosion monitoring sensor technology: (**a**) the principle of sensor technology, (**b**) the in-situ exposure experiment of sensor, (**c**) Slope of linear fit of pitting pit depth versus diameter for specimens exposed to 6 months with different levels of Cr (a_1_: 0Cr, a_2_: 0.76Cr, a_3_: 1.69Cr, a_4_: 2.52Cr), (**d**) F-index trend of alloy corrosion resistance [144].

**Figure 6 materials-16-03396-f006:**
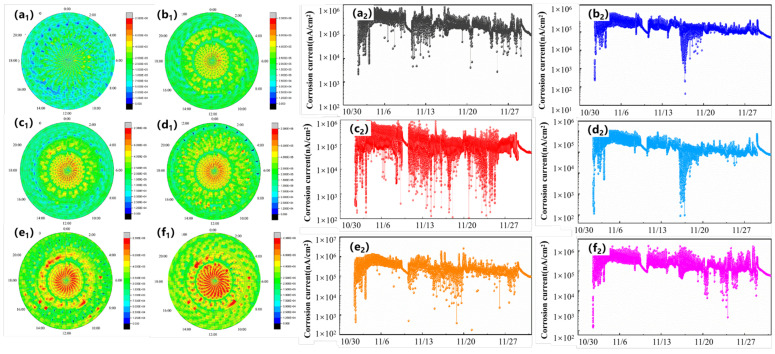
Electric coupling current densities measured by corrosion sensors for steels with different structures, (**a_1,_a_2_**) 325 °C, (**b_1_**,**b_2_**) 350 °C, (**c_1_**,**c_2_**) 375 °C, (**d_1_**,**d_2_**) 400 °C, (**e_1_**,**e_2_**) 425 °C, (**f_1_**,**f_2_**) 450 °C [160].

## Data Availability

Not applicable.

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
