# Peer review of "A Review of Trends in Corrosion-Resistant Structural Steels Research—From Theoretical Simulation to Data-Driven Directions"

_materials, 2023, doi:10.3390/ma16093396_

Round 1

Reviewer 1 Report

The authors provided an interesting review article on material corrosion modeling. However, the current version is not well-written. Hence, a major revision is required, as follows:

-          Abstract: Please change the abstract and mention that it is a review study and no additional results were added to the literature.

-          Page 1, Line 26: please remove “generally speaking”. The authors need to check the English proficiency throughout the manuscript.

-          Page 1, Lines 29-30: what do you mean grain? Please explain this phrase in the text first. Also, please use a reference for this sentence.

-          Page 1, Line 39: use “Accordingly” instead of “therefore”.

-          Page 1, Line 44: use “regarding” instead of “as” in the text. The authors should check the English of the manuscript.

-          Page 3, Line 90: please modify the text.

-          Page 10, Lines 403-406: Please do not use short paragraphs throughout the manuscript.

-          General comments: please put the phrase “review” in the title of the manuscript.

-          The quality of the figures needs to be improved. Please modify them.

-          Figure 5: the reference?

-          Please add a unique table at the end of the manuscript (before the conclusion) to determine the current research gaps needed for future studies in this field.

-          More details regarding the subjects discussed in the manuscript should be mentioned in a separate table, such as tests and methods used in different research.

-          Please rewrite the conclusion and provide key results of previous studies in simple sentences.

-          At least, one comparison figure is needed in this review article so that can help readers to distinguish the methods used for material corrosion modeling. Explaining the research in the text is not the only way of reviewing previous studies.

Reviewer 2 Report

-- The advantages of the proposed component over existing products is not clear. So, the significance of this research is not easy to judge.

-- The feasibility of such component in structural engineering is not clear. It is better to include examples of such component in structures.

-- Abstract is not clear, authors should modify it and write it in scientific style.

-- Introduction section should be rewrite again. All figures from introduction should be remove. Lat paragraph of introduction should be write as separate paragraph and it should be mention simple explanation of the article.

-- All sections should be have number.

-- Illustrate the application of the proposed model.

-- Please define all variables after their first appearance in the article.

-- Please refer all relations to valid references after the first appearance in the article.

-- What is assumption and limitation of the proposed materials and model?

-- What can we learn from the presented manuscript? What is the difference of the current work against other published articles?

Reviewer 3 Report

The article discusses the three main parts of a complete data-driven model for material corrosion: microscopic model research, mining model research of experimental data, and application of big data models. The Monte Carlo method can be used in the microscopic model to study the corrosion mechanism of materials. The mining model research focuses on predicting and modeling corrosion rates and life. The use of big data and machine learning has made progress in material corrosion modeling and has been applied in regulating corrosion-resistant performance of structural steels. This research provides an efficient way to study corrosion mechanisms and assess service life, and it also ensures the safety of social industry and construction. The construction of a corrosion database is important, and with the accumulation of data, the use of new computing tools and mathematical models will become essential to explore the corrosion process of steel. Material corrosion modeling may eventually replace field testing, and the development of new corrosion-resistant materials will become the main research direction.

The research presented in this paper is well-executed and the results are significant. It is suitable for publication in the Journal of Materials in the reviewer's view.

 Just correct these two parts:

 -In section 2.1, correct references [9, 0].

-Check the font size of formula No.2 

-The topic of the article needs correction. In the PDF format, after the word "research," there are two hyphens. Please rectify this error. 

-Section 4, which concerns the data-driven control of corrosion resistance performance, is a crucial part of your research and has a special relationship with the topic. Please provide more relevant and similar studies to support this section. 

-Please provide more explanation about the parameters in Fig. 5 for ease of understanding during the lecture. 

-Please indicate the reference for the affirmation in page 13, lines 544 to 549. 

-In Figure 6, you have provided four important pieces of information. Please separate them and provide a discussion for diagrams c and d. 

-Please provide more explanation for Fig. 6. The figure sizes need to be changed as the text is not legible. 

-The conclusion should relate to the information presented in the article. What is the main conclusion of your study? Please improve your conclusion accordingly. 

Reviewer 4 Report

In this paper, Authors given the insights of material corrosion modeling and the mastery of material corrosion regulars. It will be the future and new up coming area in Corrosion research.   Authors have studied the relationship  between the main influencing factors of material corrosion and corrosion rate, and  the performance regulation of corrosion-resistant structural steels based on big data technology. Authors have collected related papers and evaluated corrosion research.

I request authors to add these papers to strengthen the whole paper

Ketosulfone drug as a green corrosion inhibitor for mild steel in Acidic medium

    • Prasanna B. Matad

    • Praveen B. Mokshanatha

    • Narayana Hebbar

    • Venkatarangaiah T. Venkatesha∥

, and 

  • Harmesh Chander Tandon
  •  
Ind. Eng. Chem. Res. 2014, 53, 20, 8436–8444, Publication Date:April 25, 2014,  https://doi.org/10.1021/ie500232g Coatings 202111(8), 965; https://doi.org/10.3390/coatings11080965    

Round 2

Reviewer 1 Report

The authors appropriately improved the manuscript.